# COVID-19 Diagnosis and Classification Using Radiological Imaging and Deep Learning Techniques: A Comparative Study

**DOI:** 10.3390/diagnostics12081880

**Published:** 2022-08-03

**Authors:** Saloni Laddha, Sami Mnasri, Mansoor Alghamdi, Vijay Kumar, Manjit Kaur, Malek Alrashidi, Abdullah Almuhaimeed, Ali Alshehri, Majed Abdullah Alrowaily, Ibrahim Alkhazi

**Affiliations:** 1Computer Science and Engineering Department, National Institute of Technology Hamirpur, Hamirpur 177005, Himachal Pradesh, India; saloni1427@gmail.com (S.L.); vijaykumarchahar@gmail.com (V.K.); 2Department of Computer Science, Applied College, University of Tabuk, Tabuk 47512, Saudi Arabia; malghamdi@ut.edu.sa (M.A.); mqalrashidi@ut.edu.sa (M.A.); a.alshehri@ut.edu.sa (A.A.); malrowaily@ut.edu.sa (M.A.A.); 3Department of Computer Science, ISSAT of Gafsa, University of Gafsa, Gafsa 2112, Tunisia; 4School of Electrical Engineering and Computer Science, Gwangju Institute of Science and Technology, Gwangju 61005, Korea; manjitbhinder8@gmail.com; 5The National Centre for Genomics Technologies and Bioinformatics, King Abdulaziz City for Science and Technology, Riyadh 11442, Saudi Arabia; 6College of Computers & Information Technology, University of Tabuk, Tabuk 47512, Saudi Arabia; i.alkhazi@ut.edu.sa

**Keywords:** COVID-19, radiology, deep learning, CT scanning, chest X-rays, transfer learning

## Abstract

In December 2019, the novel coronavirus disease 2019 (COVID-19) appeared. Being highly contagious and with no effective treatment available, the only solution was to detect and isolate infected patients to further break the chain of infection. The shortage of test kits and other drawbacks of lab tests motivated researchers to build an automated diagnosis system using chest X-rays and CT scanning. The reviewed works in this study use AI coupled with the radiological image processing of raw chest X-rays and CT images to train various CNN models. They use transfer learning and numerous types of binary and multi-class classifications. The models are trained and validated on several datasets, the attributes of which are also discussed. The obtained results of various algorithms are later compared using performance metrics such as accuracy, F1 score, and AUC. Major challenges faced in this research domain are the limited availability of COVID image data and the high accuracy of the prediction of the severity of patients using deep learning compared to well-known methods of COVID-19 detection such as PCR tests. These automated detection systems using CXR technology are reliable enough to help radiologists in the initial screening and in the immediate diagnosis of infected individuals. They are preferred because of their low cost, availability, and fast results.

## 1. Introduction

Coronavirus disease (COVID-19), caused by SARS-CoV-2, is one of the biggest challenges of the 21st century. The entire world is battling against this virus, which has affected 182,302,122 persons and has taken the lives of 3,947,958 individuals worldwide as of 29 June 2021 [1]. The source of its origin still remains undiscovered. The WHO (World Health Organization) declared it as a pandemic on 11 February 2020 because of the widespread infection rate across China and other countries within a span of a few months.

It is a respiratory disease and is highly contagious in all age groups. Fever, sore throat, headache, cough, fatigue, and body pain are some of the known symptoms. The period between infection and the onset of symptoms may range from 2 to 14 days. It spreads via airborne droplet, and infection is caused by coming into contact, directly or indirectly, with infected individuals. Despite the fact that vaccines are now being developed and distributed, for countries with large populations such as India, the challenge still ongoing. It would take years to vaccinate every individual in the country twice. Until then, social distancing and the isolation of infected patients is the only preventive way to break the chain of infection. 

The most widely used diagnosis method is RT-PCR (Reverse Transcription-Polymerase Chain Reaction) tests. These testing kits are expensive and take 6 to 8 h to test a single sample. They also have high false-negative and false-positive rate due to their low sensitivity. Therefore, chest radiography consisting of chest X-rays (CXRs) and chest tomography (CT) scans can be further possible solutions for the detection of COVID-19 in early stages. The wide availability of already-installed X-ray machines and CT rooms in hospitals provides an added advantage. In this study, CXRs were preferred over CT scans to avoid CT room disinfection. Moreover, X-rays have lower ionizing radiation and are cheaper than CT scans. Studies show that COVID-19 leaves traces of some radiological signatures which can be identified in chest X-rays. However, these signatures can only be interpreted and analyzed by expert radiologists. This increases the chances of error and delays the process of COVID detection. Hence, there is a need for an automated diagnosis system that processes CXR and CT scan images and produces good COVID-19 detection results.

Clinical cases claim that ultrasound and chest CT perform better in excluding COVID-19 infection than in differentiating it from other respiratory diseases. Thoracic CT imaging is characterized by high specificity and low detection sensitivity to asymptomatic individuals. 

Indeed, instead of the known quantitative CT values, the distinction of COVID-19 from non-COVID-19 cases can be based on radiomic characteristics since the latter performs better than the classical quantitative CT with high values of precision, specificity, and sensitivity metrics [2].

Chest radiography involves transmitting X-rays through a patient’s chest which are reconstructed into medical images by transmitting them into radiation detectors. Figure 1 shows the CXR findings of an infected chest. These infected images are then examined and interpreted by expert radiologists. This manual process is prone to error and thus does not give high sensitivity. Indeed, according to [3], for several reasons, the accuracy of reports of radiologists when interpreting CXRs may not always be perfect due to some unavoidable errors. These errors are sometimes systemic and sometimes human. Artificial intelligence has proved itself worthy because of its high accuracy and prediction rates. In medical imaging, AI (artificial intelligence) is used to analyze and group similar patterns based on their characteristic features in image data. Recent studies proved that this can be used to detect COVID-19 and other similar lung diseases such as pneumonia.

The entire medical community is focused on the diagnosis, effective treatment, and containment strategies. This study focuses on advancing the technological tools and solutions with the contribution of various deep learning methodologies. Deep learning is advancing very rapidly and is capable of solving a wide variety of problems in all sectors. It is thus used in the healthcare sector to train CNN (convolutional neural network)-based models for the detection of COVID-19.

This study revolves around the detection of COVID-19 by applying deep learning techniques to CXR images. The remaining paper is organized as follows: Section 2 lists recent related works. Section 3 illustrates materials and methodologies used by researchers to produce classification results. Section 4 discusses the results with various performance metrics. Finally, the studies are concluded and summarized in Section 5.

## 2. Literature Review

Due to the required time (six to eight hours) for the traditional PCR method used to identify COVID-19 infection, the aim of different recent studies proposing image classifier systems has been to provide medical professionals with another rapid and low-cost method to identify COVID-19 and other pneumonia infections. 

CT scanning is a technique applied to symptomatic patients. This technique is conditioned by determining the necessary period, after the appearance of symptoms, for the realization of CT or PCR. Indeed, if the symptoms indicate an infection while the genetic test for the coronavirus is negative, CT scanning can be used as an additional procedure [4].

According to [4], the sensitivity of RT-PCR is lower than that of CT. However, this sensitivity is strongly proportional to the type of material and the method used to carry out the genetic tests. Hence, under certain conditions, the CT test can be properly included in the COVID-19 diagnostic guidelines.

A CT scan is an imaging diagnostic procedure that involves a combination of X-rays and computer technology to produce images. Hence, CT scanning, despite being more expensive than X-ray imaging, is more accurate and can be useful in providing more details in some cases.

Table 1 illustrates the relevant recent studies using deep learning models to detect COVID-19.

In what follows, we further investigate the studies mentioned in Table 1 to illustrate their advantages and drawbacks:

In [5], the authors introduce five transfer learning models to detect COVID-19 in lung CT scans. The study assesses the use of standard and contrast adaptive histogram equalization in lung scans. However, this study does not demonstrate the efficiency and impact of using histogram equalization methods on different learning models.

In [6], the authors establish an X-ray image dataset and suggest a pre-processing semi-automated model to pre-train deep learning models to detect COVID-19 and other diseases with known features. The model used allows noise from the X-ray images to be reduced. The experimental tests indicate that even simple network models such as VGG19 become more accurate (by 83%).

The study in [7] introduces a 2D convolution technique to classify CXR lung images to detect COVID-19. The dataset used is composed of 224 normal and COVID-19 images. Although the results found show good computational speed, they are performed in a small dataset with a limited number of features.

In [8], a ResNet-50 transfer learning model is used to classify COVID-19 CXR images. The high obtained classification accuracy (99.5%) can be used in clinical practice. However, this accuracy is obtained using a small dataset. More successful deep learning models can be used with larger datasets.

In [9], a concatenation of features extracted from two transfer learning models are used to detect COVID-19 with X-rays, CT scans, and two biomarkers. With the same results, the introduced concatenation gives a better computational time than the VGG16-ResNet50 concatenation. However, the introduced concatenation has a high number of parameters. In addition, the introduced concatenation can truly predict positive and negative cases from only two positive or negative images, respectively.

In [10], a two-dense-layer model is proposed to detect COVID-19 from CXR images. Batch normalization is introduced in the second layer to avoid the overfitting of the model. Three data types are used in the proposed dataset: normal images, pneumonia images, and COVID-19 pneumonia images. This dataset is used to assess the efficiency of the introduced model compared with Xception, Inception V3, and Resnet50 models. The introduced model has lower loss and higher accuracy than other models in both validation and training data. However, one of the drawbacks of the introduced model is that it takes considerable computational time to obtain important features. Moreover, adding layers in the model can mean training takes longer.

In [11], the authors collect a set of X-ray lung images to evaluate the severity of COVID-19 pneumonia. Bounded boxes are used to identify the diseased area. An RCNN mask can be added to the model used to enhance the accuracy of the detection of the diseased area.

The study in [12] aims to detect COVID-19 from a dataset of 2727 chest radio open-source images using different pre-trained convolutional neural networks as learning models. As a result, the VGG-16 model achieves better classification than the F1 score and gives less false positives and false negatives compared to DenseNet and VGG-19. However, the dataset used should contain more variated images from different groups of geographical regions, races, and ages. Moreover, more training models should be tested to gain a comprehensive comparative analysis between the different models.

In [13], a CSEN recognition model, combining the advantages of representation-based techniques and deep learning models, is used to detect COVID-19 pneumonia. Using training samples and a dictionary, the CSEN establishes mapping from the sparse support coefficients to the query samples. In terms of memory and speed, the proposed CSEN-based system is computationally efficient, but the main issue with it is that its performance rapidly degrades due to the scarcity of data.

In [14], a Mix-Match-based semi-supervised learning system is used to identify the positive cases of COVID-19. The advantage of such a semi-supervised system is the use of unlabeled data which are more available.

In [22], four pre-trained models are used to detect COVID-19 from a dataset composed of 5000 non-COVID X-ray images and 200 COVID X-ray images. However, a larger set of labeled COVID-19 images is needed to accurately estimate the performance of the tested models.

The authors of [23] address the problem of the classification and recognition of COVID-19 images using different CNN pre-trained models. The study concludes that ResNet-34 is better than other networks. However, this study relies on a binary classification (normal or infected by COVID-19) and does not address a multi-class classification for more detection accuracy.

The study in [15] aims to identify COVID-19 using a feature fusion deep learning model. Using K-fold cross-validation tests, the efficiency of the latter introduced model is confirmed to be more accurate than other classification methods (CNN, SVM, KNN, and ANN). The idea of the method is to combine the features extracted by CNNs and those extracted by histogram-oriented gradients (HOGs). The study deduces that choosing appropriate classification and selection features is necessary for COVID-19 detection from X-ray images.

In [16], the authors use a local dataset of X-ray and CT images. The introduced method, based on a deep learning model, achieves good results, but the dataset used is relatively small.

The authors of [17] suggest a three-class (namely: normal, pneumonia, and COVID-19) classification model to detect and classify X-ray images. A transfer learning InceptionV3 model is used. Additional layers are added to enhance the model. The experimental tests indicate the high performance of the three used classes.

In [18], the authors propose a deep multi-layer neural network system named nCOVnet to detect the presence of COVID-19 from X-ray images. Despite the reported high accuracy of the introduced system, the latter relies on a limited training set.

The study in [19] introduces an (L, 2) transfer learning system to classify COVID-19 CT images. The measuring indicator used relies on a micro-averaged F1 score. Despite the fact that the results show that the proposed system is efficient in detecting COVID-19 images, it has some drawbacks: data from different sources such as CT data combined with CXR and historical data are not easily handled. In addition, the used dataset is not clinically verified and is category-limited.

In [20], a three-classes-deep LSTM system is introduced to detect COVID-19 from MCWS images. The dataset used is public, and the comparison with the introduced system with other learning methods gives excellent results. According to the authors, the accuracy of the results can reach 100%.

The study in [21] suggests a DNN X-ray image classifier. Four classes (COVID-19, bacterial, viral, and healthy) and seven scenarios are considered. The results are promising compared to other deep transfer learning systems such as MobileNet, VGG16, and InceptionV4. The resistance of the introduced model to the noise of images is good.

To sum up, the main drawbacks of the previous investigations and recent studies regarding the classification and detection of COVID-19 from images are:The absence of scalability evaluation, which is important in estimating the model performance under real-world operation settings.Complexity and statistical tests are not given to investigate the robustness of the results.The small sizes of datasets and non-variability of data in datasets.The absence of the consideration of multi-class classification: only binary (normal vs. infected) classifications are taken into consideration in most studies.The absence of hybridizations of deep learning models with other techniques such as evolutionary multi-objective optimization for more accuracy in detection.

Further investigations and discussions of the interest in the use of learning models for COVID-19 detection from images have been given in recent surveys [24,25].

## 3. Materials and Methods

Machine learning, like optimization and other artificial intelligence methods, has been proven to be very useful in resolving real-word complex problems related to engineering issues [26,27,28,29] or medical ones [30,31], as in the case of COVID-19 detection. Figure 2 illustrates the principally used machine learning techniques for CXRs.

To understand and study the relationship between CXR and other deep learning frameworks for COVID-19 diagnosis, this paper reviews various publications and research articles published from March 2020 onwards. The sources used included ScienceDirect, Google Scholar, ArXiv, IEEE, Springer, ACM, etc. and some of the keywords used for the search were “coronaviruses”, “COVID-19 Diagnosis”, “Deep Learning”, “transfer learning”, “Chest Radiography”, and “CNN”. While this study mostly focuses on diagnosis using CXR images, some overlapping techniques used for diagnosis based on chest CT images were also considered. The majority of the research works used deep transfer learning on the ImageNet dataset. CNN architectures were trained with the different tuning of their hyperparameters. The following subsections provide an overview of various state-of-the-art approaches and datasets used to review this survey.

### 3.1. Types of Classification

The COVID-19 detection task was carried out by classifying the X-ray images using either binary classification, i.e., 2 classes or multi-class classification, i.e., 3 or 4 classes. Each class was labeled by one of the following—“COVID-19”, “healthy”, “no-findings”, “viral pneumonia”, or “bacterial pneumonia”. The binary classification consisted of “COVID-19” as one of the classes, and the other class could be either of the other four, i.e., “non-COVID”. The three-class classification labels were “COVID-19”, “pneumonia”, and “no-findings”. Most studies used binary or triple-class classification. However, some other studies suggested a classification into four classes: “COVID-19”, “viral pneumonia”, “bacterial pneumonia”, and “no-findings or healthy”. Indeed, binary classification represents the dichotomization of a practical situation of a problem using classification rule to decompose the elements of a set into two classes (groups). On the other hand, if there are more than two classes, the classification process is qualified as multi-class classification [32]. It is worth mentioning that binary classification may be customized in several ways to handle multiple classes [33].

Among the issues of binary classification, there is a limited number of classes where only two values for the outcomes are possible: “yes” or “no”. The binary classification can misinterpret the patient’s infections which can lead to errors such as false negative and false positive. False negative occurs if an infected person is categorized as healthy. False positive occurs if a healthy person is categorized as infected.

Among the issues of multi-class classification is the problem of imbalanced datasets. Imbalanced data indicates a problem with a set of inequal representations of classes. The inequal repartition of data can lead to the lower performance of conventional machine learning techniques in the prediction of minority classes. Indeed, multi-class problems with imbalanced datasets are more challenging than binary problems with imbalanced datasets. 

Regarding the studies completed, most of them are interested in the binary classification of COVID-19 [34,35,36], and very few studies suggest the multi-class classification of COVID-19 [37,38]. Actually, the performance of multi-class techniques should be improved more.

### 3.2. Deep Learning Architectures

The operation of deep learning architectures can be explained as follows: One of the deep learning techniques is artificial neural networks, which involve massive amounts of data for computation. This type of learning automatically learns from instances of data. One of the classic architectures of convolutional neural networks is the VGG. To increase the depth of the network layers, VGG, for example, uses a filter analyzer, a connected layer, and a set of shared layers.

Deep transfer learning and CNNs are widely used in medical imaging applications. Transfer learning is used for those models where the training set is inadequate and training the model from scratch is not feasible. In transfer learning, pre-trained networks are used with the fine-tuning of parameters that performed other traditionally trained networks from scratch for COVID-19 diagnosis. Figure 3 contains a block diagram representing the steps involved in classifying COVID-19 cases from CXR using transfer learning deep CNN architectures.

CNN architectures consist of convolution layers and other pooling layers. They perform well in classification tasks related to computer vision and hence also in assessing medical images. In the further subsections, various CNN architectures and their methodologies are reviewed to identify COVID-19 patients from raw X-ray images.

#### 3.2.1. VGG

This neural network performed very well in the ImagNet Large Scale Visual Recognition Challenge (ILSVRC) in 2014 and was proposed by the Visual Geometry Group (VGG). It mainly consists of 16 or 19 convolution layers and is capable of achieving good accuracy.

VGG16 is a classification and detection technique widely used in the field of image processing. Known for its ease of use for transfer learning, VGG16 can efficiently classify (with an accuracy up to 92.7%) a thousand of images having distinct types.

As far as VGG-19 is concerned, it is 19 layers of depth brought together in a convolutional neural network. In the same previous imaging application context, a database called ImageNet [39] contains over a million images and uses VGG-19 to provide a pre-trained version of the network that can differentiate a thousand types of image objects.

However, due to the large width of convolution layers, its deployment has high computational requirements both in terms of time and memory. For CXR images, VGG-16 extracts features at a low level due to its small kernel size. An attention-based VGG-16 model [40] proposed four main modules—an attention module, convolution module, fully connected layers, and Softmax classifier.

This was implemented using a fine-tuning approach on other pre-trained networks. Three different datasets were used to train the model—D1 for triple-class, D2 for four-class, and D3 for five-class classification. The classification accuracy obtained for 18 parameters was 79.58%, 85.43%, and 87.49%, respectively.

Rahaman [41] evaluated the VGG-19 model and obtained the highest testing accuracy of 89.3% among other CNN architectures. Another study in [42] used the VGG-16 and VGG-19 models for the feature extraction of COVID-19 with SVM with 92.7% and 92.9% accuracy, respectively.

#### 3.2.2. GoogleNet

GoogleNet or Inception V1 [43] was the winner of the ILSVRC 2014 image classification challenge and has a lower error rate than VGG. This consists of 1 × 1 convolution, an inception module (IM), and a global average pooling. The convolution size is the same in each layer. These inception modules learn spatial correlations and cross-channel correlations. The IM reduces dimensionality as its output is smaller than the input in terms of feature maps. Moreover, significantly deeper models could be trained using IM by reducing the trainable parameters by up to 10 times. Other variations of GoogleNet such as Inception V2, Inception-ResNet, Inception V3 [44], and Inception V4 [45] have been developed by slightly varying the inception module.

The authors of [46] performed binary and multi-class classification using GoogleNet and achieved an accuracy of 98.15% for the binary and 75.51% for the multi-class classification of COVID-19 cases. Similarly, the authors of [42] also used GoogleNet with an SVM classifier to obtain an accuracy of 93%.

#### 3.2.3. AlexNet

This architecture requires less training time and fewer eras compared to other previously trained transfer learning models. It also gives outstanding results in the recognition and classification of images. This network is also known to give the highest accuracy on the ImageNet dataset. The network consists of 5 convolutions, 2 hidden, and 1 fully connected layer, making the depth size 8. The input image size was 227 × 227 with 61 million parameters fine-tuned. The dropout method was used to deal with overfitting, and it enabled the network to learn more features. The ReLU activation function was used.

The study in [47] reduced the original 1000 classes in AlexNet to 3 classes—COVID-19, normal, and abnormal. The model was trained on three sets of datasets obtained from various open-source networks and radiological society websites. Researchers further modified the original architecture and proposed four effective AlexNet models that detected and classified CXR images accurately [48]. In [46], the obtained accuracy was 97.04% in binary classification and 63.27% for multi-class classification using AlexNet. The study in [35] used AlexNet for feature extraction and achieved 95.12% accuracy for three classes—COVID-19, SARS, and normal—in their project DeTraC. Each class was divided into separate subclasses and reassembled to give the final prediction outcomes.

#### 3.2.4. MobileNet

This architecture uses separate convolutions depthwise to create lightweight neural networks for embedded and mobile system applications. Balance is maintained by a tradeoff between the hyperparameters for accuracy and latency. It has 53 layers and about 3.4 million trainable parameters [49]. It consists of depthwise convolutions, expansion, and projection convolutions.

For COVID-19 detection, MobileNet was used to achieve accuracies of 60% [50] and 96.30% [46]. However, for distinguishing COVID-19 from normal cases, the mean accuracy using MobileNet-V2 was 87.61%, and for COVID-19 and pneumonia, the mean accuracy was 97.87%. For three-class experiments, it resulted in 92.85% accuracy.

#### 3.2.5. ResNet

On increasing the depth of the network in CNNs, the training error increases. ResNet solves this problem by introducing a residual unit. ResNet-18 and ResNet-34 consist of two deep layers, and ResNet-50/101/152 has three deep layers. The residual learning component reuses the activation from previous layers and skips the layers that do not contribute to the solution. It uses batch normalization and identity connection to mitigate the vanishing gradient problem and improve performance.

It was observed that ResNet is one of the most widely used CNNs in COVID diagnosis studies [51]. The authors of [52] used three variants of this—ResNet-18 for 91% accuracy, ResNet-50 for 95%, and ResNet-101 for 89.2% classification accuracy. ResNet-18 and ResNet-50 were also trained on an imbalanced dataset that had 3000 normal and 100 COVID-positive CXR images. A 89.2% detection accuracy rate in ResNet-50 and a 98% sensitivity rate were obtained in both variants [22].

#### 3.2.6. Xception

Xception [53] stands for an extreme version of Inception, and it also uses depthwise separable convolutions like ResNet instead of traditional convolution. It outperformed Inception V3 on the ImageNet dataset of 17,000 classes and 350 million images. It enables the learning of spatial patterns and cross channels separately. The depthwise separable convolutions lower the number of operations and hence the computational cost by a huge factor.

Inception V3 gave an accuracy of 78.2%, which was increased to 79% on the ImageNet dataset. The authors of [50] diagnosed COVID-19 using the Xception model, and it resulted in the highest precision in detecting COVID-positive cases among the rest of the deep learning classifiers. However, it did not perform well in classifying normal cases. It resulted in a sensitivity rate of 0.894 and 0.830 precision.

CoroNet [38] is another CNN model based on the Xception architecture with two fully connected layers and a dropout layer at the end. It has 33,969,964 trainable parameters. Four-class, three-class, and two-class variants of the CoroNet model were proposed and pre-trained on the ImageNet dataset. In [49], an optimizer with a batch size of 10 and 80 epochs was used for re-training. The mean accuracy values were 89.6%, 95%, and 99% for 4-class, 3-class, and 2-class, respectively.

#### 3.2.7. DenseNet

DenseNet is somewhat similar to ResNet with a few differences. It connects the previous layer to the forward layer by concatenation. Therefore, in a network of *n* layers, it has *n*(*n* + 1)/2 connections. DenseNet is more efficient than other state-of-the-art CNN architectures such as ResNet in image classification parameters and computational terms. The convolution in the network generates fewer feature maps as the layers are densely connected. Redundancy is lower as layers reuse the features and propagate them.

DenseNet was built to resolve the vanishing gradient problem in neural networks, that is, the loss of information before reaching the final output layer because of longer paths. The different versions of DenseNet based on the number of layers computed are DenseNet-121, DenseNet-160, and DenseNet-201. This was the second most used architecture in the previously reviewed studies.

The study in [50] used DenseNet-201 to achieve an accuracy of 90% which was later improved in [42] to 93.8%. A specificity rate of 75.1% was obtained in another application of DenseNet [22]. The authors of [46] used this to achieve a multi-class classification accuracy of 93.46% and binary classification accuracy of 98.75% using DenseNet. DenseNet is also used as a backbone in developing other COVID-19 diagnosis systems using chest CT [54].

#### 3.2.8. SENet

In 2017, the authors of [55] proposed the Squeeze and Excitation Network, which was also the winner of the ILSVRC Challenge 2017. It reduced the top-5 error rate to 2.251% and surpassed the winning entry of the previous year. It was based on the relationship and interdependencies between the channels in a convolution network. It introduces an additional computation known as the SE block and integrates it with other CNNs such as ResNet. This block is added to every residual unit in ResNet to improve performance. This new merger is called SE-Inception-ResNet-v2 and SE-ResNet-50. Though this resulted in increased complexity in computation, it yielded consistent good returns compared to increasing the depths of ResNet architectures. Experiments with non-residual networks such as VGG were also conducted, which also resulted in improved performance.

The SE block is the main building component which comprises three layers—the dense layer, squeeze dense layer, and global average pooling layer. The SE block emphasizes cross-channel patterns rather than spatial patterns and it learns the image objects that are bundled together. The output of the block retains the essential features and downscales the irrelevant feature maps.

The authors of [46] used SENet for the binary classification and multi-class classification of COVID-19 cases and obtained an accuracy of 98.89% and 94.39%, respectively. The study in [22] used this in place of ResNet and obtained 98% sensitivity and 92.9% specificity. The dataset used was highly imbalanced, and the specificity and sensitivity values for ResNet were 89.6% and 90.7%, respectively.

#### 3.2.9. ShuffleNet

Other CNN architectures used were CapsNet, autoencoder, and ShuffleNet [56]. For the ImageNet classification task, these performed better than MobileNet. ShuffleNet was approximately 13 times faster than AlexNet with comparable accuracy values. It has pointwise group convolution operations and channel shuffles to reduce the computations. This enables the flow of information across various channels.

The authors of [57] used feature extraction automatically which was then given to different classifiers—KNN, random forest, SVM, and Softmax. The accuracies obtained with these four classifiers were 99.35%, 80%, 95.81%, and 99.35%, respectively.

#### 3.2.10. DarkCovidNet

DarkNet-19 is the model that is based on a real-time object detection system—YOLO (You only look once) [58]. Rather than designing the entire model from scratch, it is picked as the starting point. The successful architecture of the DarkNet classifier makes it more efficient. Fewer layers and different filter sizes were used compared with the original DarkNet architectures [59]. Filters were gradually increased from 8 to 16 to 32.

DarkNet-19 comprises 19 layers of convolution and five of pooling—Maxpool. These layers are standard CNN layers with varying sizes, filter numbers, and stride parameters. 

The modified DarkCovidNet layout has 21 convolution layers and 6 pooling layers. This is a modification of the DarkNet model. It consists of 21 convolution layers. Each DarkNet (DN) layer has one convolution layer with a block size = 3 and stride value = 1. It is followed by batch normalization and the Leaky ReLU activation function. Each triple convolution (tri Conv.) consists of three similar sequential DarkNet layers. The batch normalization is used to reduce the training time and stabilize the model. Leaky ReLU is used as a modification of ReLU with a negative slope of 0.1. Other activation functions such as sigmoid and ReLU could also be used, but they give zero values in their negative side of derivatives. Leaky ReLU overcomes this problem of vanishing gradients and dying neurons. The optimizer in [49] is used to update the weights and loss entropy functions, and the learning rate is taken as 1 × 10^−3^. This model was evaluated using various performance metrics and resulted in 97.6% binary classification accuracy and 88% triple-class classification accuracy.

### 3.3. Comparing the Binary and Multi-Class Classification for COVID-19 Detection

So, which type of classification is more suitable for COVID-19 detection?

The study in [60] suggests diagnosing COVID-19 using a system called ECG-BiCoNet which combines deep bi-layers and ECG data to distinguish COVID-19 cases from other cardiac ones. Both binary and multi-class classification were proposed for the ECG-BiCoNet with an accuracy of 98.8% and 91.73%, respectively. The results for numerous classifiers such as RF, SVM, and LDA indicate that:-The binary classification detects the cardiac variations in ECG images caused by COVID-19 and differentiates it from healthy ECG images. However, binary classification increased the computation load of the training models.-The multi-class classification properly achieved the detection of COVID-19 cases. However, it has less ability, compared to binary classification, to detect other cardiac diseases and normal ECG images. The computation cost is slightly enhanced compared to the binary classification.

Another study [61] suggested a binary classification (COVID-19 infected or healthy) and multiple classification (pneumonia, COVID-19 infection, or healthy). A 17-layered CNN model with numerous sizes of filters was proposed for the training of CXR images. The accuracy of the model was 98.08% (87.02%, respectively) for binary classification (multi-class, respectively).

## 4. Results and Discussion

The previous section lists some of the famous works and CNN architectures proposed for the detection of the COVID-19 virus from CXR images. This section provides in-depth analysis and insights into the studies reviewed.

### 4.1. Datasets

The articles reviewed experimented with 13 different datasets. Table 2 summarizes these datasets with their respective names, the number of images, the resolution of each image, and references. Most of the images are in JPG, JPEG, or PNG format. The Cohen Image Collection [62] is known to be the most cited dataset and was used in almost 85% of the works. This may have resulted in lower image quality as it was collected from online publications rather than original medical reports. It consists of images obtained from various websites and online publications which could assist researchers in developing AI-based deep learning models. The Cohen Image Collection [62], accessible from [63], is a dataset that defines the first initiative to collect clinical cases and public data for COVID-19 as images. Representing the largest prognostic dataset on COVID-19, this dataset involves hundreds of X-ray images in frontal views. It is a reference for the development of decision support and machine learning systems via COVID-19 image processing. The aim of such systems is the prediction of patient survival and the interpretation of their disease development cycle. The images in the Cohen dataset, in both lateral and frontal views, reflect metadata such as survival status, incubation status, time to onset of initial symptoms, and hospital location.

The COVID-19 Image Data Collection is named the “Montreal database” in some of the studies. The COVIDx dataset consists of 48 COVID images and is updated constantly [64]. The COVID-19 Dataset Award was won by the COVID-19 Radiography Database, which is a composition of different datasets: the SIRM (Italian Society of Medical and Interventional Radiology) Database, Twitter COVID-19 CXR Dataset, RSNA Pneumonia Detection Challenge Dataset, Kaggle, and other online sources.

The Open-I repository is an open-access biomedical search engine maintained by the US National Library of Medicine in which CXR images can be found with the relevant publication [76]. The Twitter CXR Dataset [72] was shared by a cardiothoracic radiologist in Spain on his Twitter account and it consists of 135 images having SARS-Cov-2 viral infection. The CXR-8 [68] contains frontal-view CXR images of more than 30,000 patients infected with 14 thoracic diseases. This is also known as the RSNA Pneumonia Detection Challenge dataset. Most of these studies combined datasets to increase the training data so redundant data can also be found. The above-stated datasets are publicly available from various online repositories and platforms. However, the rest are obtained from local hospitals. The latter datasets cannot be accessed publicly. GitHub and Kaggle are the most used portals to store and access these datasets.

### 4.2. Performance Comparison

Despite numerous studies using deep learning models on various datasets, identifying the most efficient architecture is still a difficult task. The variation in testing and training data also added to the differences and complications in comparing the CNN models on standard performance metrics. Most works evaluated their models based on accuracy, F1 scores, specificity and sensitivity rates, the area under the ROC curve, and Kappa statistics.

The differences between the COVID-19 datasets are still unresolved. Thus, a standard COVID-19 dataset should be maintained by the research community with which every researcher could validate their respective models. Standard evaluation metrics should also be specified to ease the comparison among them and test their efficacy. Table 3 contains the summary of results in terms of mean accuracy, F1 score, and AUC score. Specificity, sensitivity, precision, and recall could also be taken for comparison, but they are bound to differ for two-class, three-class, and four-class classification.

What follows is the signification of the metrics used:

-Accuracy is the most natural measure of performance. It is defined by the percentage of correct predictions compared to the total number of observations. The efficiency of the model is then proportionally linked to the value of its accuracy. In general, an accuracy A is defined by A = TP + TN/TP + FP + FN + TN (knowing that TP = true positives, TN = true negatives, FP = false positives and FN = false negatives). A model having A = 0.76, for example, indicates that this model is approximately 76% accurate. On the other hand, the precision effectively measures the performance only if the data are symmetric (values of the FN are comparable to those of the FP). Hence, other performance metrics should be tested alongside the accuracy. -Precision: This metric defines the relationship between the total number of predicted positive observations and correctly predicted positive predictions. A typical example of using this metric is, in a disaster, how many people actually survived among those described as having survived? The good performance of this metric is inversely related to the rate of false positives. The precision formula is generally described by TP/TP + FP.-Sensitivity (recall) is another metric that describes the relationship between the real actual class observations and the correctly predicted positive observations. Sensitivity tries to answer the question: how many passengers did we tagged out of all the passengers who actually survived? Sensitivity is usually defined by the formula TP/TP + FN.-F1 score is a metric that reflects the weighted average value of sensitivity and precision. This means that the F1 score considers false negatives and false positives simultaneously. In the case where the cost of false negatives is very different from false positives, we must use precision and recall at the same time. If the class distribution is unequal, the F1 score is more useful than the precision. The latter performs better if we have a similar cost of false negatives and false positives. The general formula of the F1 score is as follows: 2 × (Precision × Sensitivity)/(Precision + Sensitivity) [84].

The CXR image databases are used in [85,86], and the accuracy is presented as 98.70%, 88.80%, 95.70%, and 99.90%, respectively.

### 4.3. Class Imbalance Problem

The major challenge faced in studies on COVID-19 is the limited availability of COVID-positive image datasets. As is evident from the above statistics, the number of COVID-19 images is very small compared to the normal and pneumonia classes. This uneven distribution leads to the class imbalance problem. Some studies focus on data augmentation to enlarge the COVID dataset [87]. Another solution is to take an equal number of images in each class. However, deep models such as ResNet do not perform well with a lower amount of training data.

A study proposed the use of the SMOTE (Synthetic Minority Oversampling Technique) which is a kind of data augmentation technique for minority classes [52].

The number of COVID images varies widely in numbers compared to total data samples. Some studies were simulated with as few as 11, and others took as many as 1536 COVID-19 images, whereas the total number of images was in the range of 50 to 224,316. Thus, AI researchers used different techniques to tackle this problem [88]. The authors of [57] fixed the number of image samples to 310 in each class. The authors of [46,51] also used a fixed number of samples in each class. Both studies [82,89] used a class-weighted entropy loss function. Others emphasized cost-sensitive learning.

### 4.4. COVID-19 Severity Prediction

Another challenge faced in containing this pandemic is the inability to predict the severity of a patient diagnosed as COVID-positive. Based on symptom period analysis and past CXR records, researchers are working to predict the severity of patients in terms of COVID score.

Assessing the progression of the disease and its effect on the lungs could identify patients at high risk. They could then be treated with extra attention and care from the medical personnel. Deep learning techniques developed on CXR images of patients could assist doctors in tracking, assessing, and monitoring severity and progress and hence aid in efficiently triaging patients. One study [89] monitored patients and predicted whether their condition would improve or worsen in the coming days with an accuracy of 82.7%. The more deadly L- and H-type strains were also identified using DL architectures. Categorizing multiple scans of the same patient, extracting features using DL, and using embedded machine learning algorithms on these features were also used to monitor the condition and recovery of patients. GANs (Generative Adversarial Networks) have also given promising results in severity prediction.

## 5. Conclusions and Future Scope

This study investigated the diagnosis of COVID-19 cases using AI and deep learning techniques with CXR images given as input. An automated diagnosis system is needed to overcome the shortage of testing kits and speed up the screening process with the limited involvement of medical professionals. Numerous deep learning architectures proposed by researchers were reviewed and discussed.

The proposed models were validated on different datasets, especially by Cohen Image Collection data, which is the most cited dataset. The attributes and descriptions of different datasets used were discussed. The models were later evaluated and compared based on performance metrics such as accuracy, F1 score, and AUC values. However, due to limited instances and the availability of COVID-19 CXR images, almost all datasets are highly imbalanced. So, classification accuracy cannot be the only metric to evaluate and compare the efficiency of these models. Additionally, less training time, a reduced error rate, and good performance with the limited amount of training data are also considered important in developing these models.

Transfer learning produced improved results as the models are pre-trained rather than built from scratch. Different studies used several numbers of classes to identify COVID-19 cases. Binary classification consisted of COVID-19 and normal classes, whereas multi-class classification was further divided as three-class, four-class, and five-class classification. Viral and bacterial pneumonia cases were also segregated along with COVID-19 and no-findings classes.

These proposed methodologies need to be validated on larger datasets with specified standards and evaluation metrics before they come to practice. AI researchers should work closely with expert radiologists to analyze the results and find a tradeoff between the deep features learned automatically and the features extracted by domain knowledge to obtain an accurate diagnosis. Additionally, most of the reviewed works used data augmentation to overcome the problem of a lack of COVID data. GAN network implementation could be used to generate new data as well as predict the severity of patients based on symptom period analysis.

The detection of COVID-19 using radio images and deep learning techniques seems to be a very promising method because of the actual issues encountered by the vaccines regarding the non-acceptation of vaccination by people and regarding the newly appearing COVID-19 variants threatening the efficiency of vaccines.

These techniques could also be used to detect other chest-related illnesses such as pneumonia, tuberculosis, etc., in the near future. More diverse datasets could be used to increase the robustness and accuracy of the model. Mobile applications could be developed via the cloud to assist the initial screening of patients. Radiological screening for COVID-19 diagnosis is an active research area, and sooner or later, the medical community will have to rely on these methods as the pandemic progresses. Furthermore, deep learning techniques can be combined with other intelligent AI methods such as optimization algorithms [90,91] for a better manipulation of radiological imaging for COVID-19 detection. 

## Figures and Tables

**Figure 1 diagnostics-12-01880-f001:**
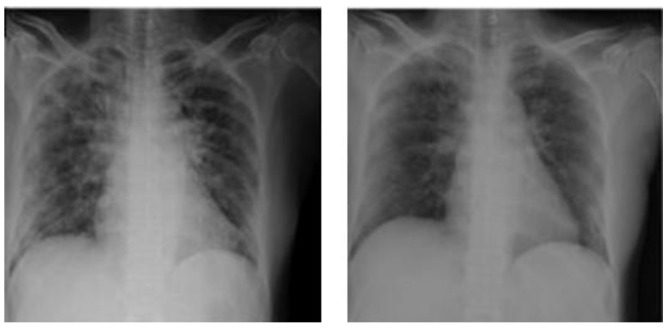
Ground glass opacities in CXR findings.

**Figure 2 diagnostics-12-01880-f002:**
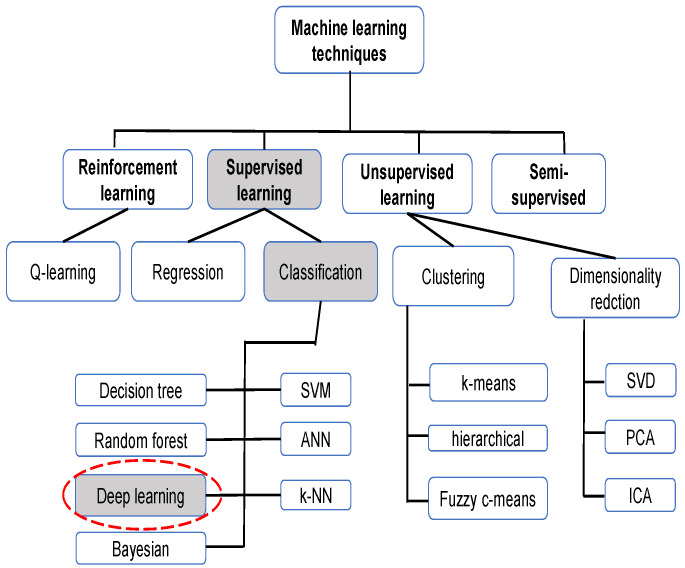
Machine learning techniques for CXR.

**Figure 3 diagnostics-12-01880-f003:**
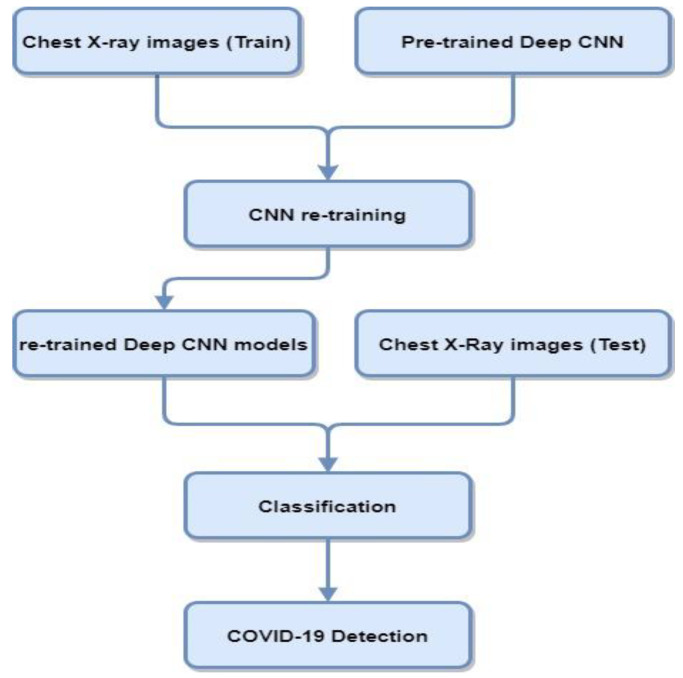
Block diagram of deep CNN architectures using CXR for COVID-19 detection.

**Table 1 diagnostics-12-01880-t001:** Recent studies proposing learning model for classifying images to detect COVID-19.

Study	Application	Learning Method	Model	Features	Dataset Specifications			Results (Accuracy, Specificity, Sensitivity)
					Accessibility	Type/Structure	Size Structure	
(Lawton and Viriri, 2021) [5]	CT lung scans	Different transfer learning architectures: DenseNet-201, ResNet-101, VGG-19, fficientNet-B4, and MobileNet-V2	Fully connected artificial neural network: single 256-node hidden layer using the ReLU activation function and a two-node softmax output layer.	Peripheral and bilateral predominant ground-glass opacities	Publicly available		2482 images of São Paulo patients: 1252 COVID-19-positive and 1230 normal	Best performance is on VGG-19: accuracy = 95.75% recall = 97.13% F1 score of 95.75, and ROC-AUC of 99.30%. Specificity = 94.42%
(Horry et al., 2021) [6]	X-ray-image-based COVID-19 detection	Convolutional neural network models	VGG, Inception, Xception, and Resnet	Patchy infiltration or opacities	Publicly available	X-ray images	200 × normal vs.100 × COVID-19100 × pneumonia	Both VGG16 and VGG19 classifiers provided good results within the experimental constraints of the small number of X-ray images. Around 80% for both recalls, and simpler networks such as VGG19 performs relatively better with up to 83% precision.
(Padma et al., 2020) [7]	Illustrates the severity of coronavirus in lung using radiology images	Convolution 2D technique		Binary classes	Open-source datasets of COVID-19 available at GitHub and Kaggle	Images of Chest X-ray and CT scan	60 images where 30 are normal and 30 COVID-positive images	Accuracy for training set of 99.2%, validation accuracy of 98.3%, loss 0.3%, sensitivity of 99.1%, specificity of 98.8%, and precision of 100%
(Karhan et al., 2020) [8]	Radiological images of the chest	Convolutional neural networks	ResNet-18 (Residual Network)	Two different classes in the dataset (COVID-19 and non-COVID-19)	Hybrid: Italian Society of Medical Radiology (SIRM) dataset, coronavirus open-source shared dataset, and dataset created by compiling diagnosed images	COVID-19-positive and -negative CXR images		Accuracy rate of 99.5%
(Hilmizen et al., 2020) [9]	Diagnosing COVID-19 pneumonia from chest CT scan and X-ray images	Multimodal deep learning: concatenation of DenseNet121-MobileNet	The input data for the feed of the network were normalized, resized to 150 × 150 pixels, and the number of channels was set to 3 (RGB images).	Two open-source datasets, and the allocation for each class was balanced	Public		1750 data for each dataset in the trainingset that has 750 data on the validation set for each dataset	The concatenation of DenseNet121-MobileNet gives accuracy of 99.87%, sensitivity of 99.74%, and specificity of 100%
(Santoso and Purnomo, 2020) [10]	COVID-19 detection based on the CXR images	Deep neural network	Modification of deep neural network based on Xception model			618 images with 256 × 256 in size. The data are categorized into normal people, pneumonia and pneumonia caused by COVID-19	The dataset is divided into data training (72.3%), data validation (18.0%), and data testing (7.7%)	Xception accuracy of 90.09 and loss of 0.6458
(Darapaneni et al., 2020) [11]	Analysis of severity of COVID-19 from chest X-ray images	Segmentation mask prediction	Mask RCNN				4668 trained images and 1500 tested images	Mean average precision is 90 (89,) on training set (test set)
(Kandhari et al., 2021) [12]	Detecting COVID-19 from CXR	Deep learning models	ResNet, DenseNet, VGG 16 and VGG 19		Open source		Dataset of 2727 images	VGG-16 model achievedan impressive classification accuracy of 98.9% and F1 score of 0.984 with high sensitivity and specificity as well
(Yamac et al., 2021) [13]	COVID-19 recognition approach directly from X-ray images	Convolution support estimation network (CSEN)	CSEN that can be seen as a bridge between deep learning models and representation-based methods		Different publicly available datasets		A benchmark X-ray dataset, namely QaTa-Cov19, containing over 6200 X-ray images is created. The dataset covers 462 X-ray images from COVID-19 patients along with three other classes; bacterial pneumonia, viral pneumonia, and normal	Over 98% sensitivity and over 95% specificity
(Calderon-Ramirez et al., 2021) [14]	Scarce labeled data classification	CNN models	Semi-supervised deep learning with Mix Match					Accuracies higher than 90%
(Alam et al., 2021) [15]	Earlier detection of the COVID-19 through accurate diagnosis	CNN (VGGNet)	Feature fusion using histogram-oriented gradient (HOG) and convolutional neural network (CNN)				Normal (1900) images. The confusion metrics of the generalization results are presented in Fi	Testing accuracy of 99.49%, specificity of 95.7%, and sensitivity of 93.65%. (98.36%) provided higher accuracy than the individual feature extraction methods, such as HOG (87.34%) or CNN (93.64%)
(Gilanie et al., 2021) [16]	Coronavirus (COVID-19) detection from chest radiology images	Convolutional neural networks			Three publicly available and a locally developed dataset, obtained from Department of Radiology (Diagnostics), Bahawal Victoria Hospital, Bahawalpur (BVHB), Pakistan			The proposed method achieved average accuracy of 96.68%, specificity of 95.65%, and sensitivity of 96.24%.
(Amin et al., 2021) [17]	Classification of COVID-19 X-ray images	Deep learning model	Transfer Learning InceptionV3	Three different classes: pneumonia, normal, and COVID-19	Public CXR images (pneumonia) from Kaggle + COVID-19 images dataset from GitHub	299 × 299 pixels	194 images from the original pneumonia dataset + 163 images from the COVID-19 dataset	98% accuracy, precision and recall. F1 scores all are equal to 0.97 for pneumonia and normal images and are equal to 1 in COVID-19 images
(Panwar et al., 2020) [18]	Fast detection of COVID-19 in X-rays	Deep learning	Transfer learning model with five different layers	VGG as a model for feature extraction	Public	All images converted to a standard size (224 × 224 pixels)	142 for COVID-19 images; 5863 Kaggle CXR healthy images	97.62% of true positive rate; accuracy up to 99%
(Wang et al., 2020) [19]	COVID-19 detection in chest CT images	Multiple-way data augmentation	Offline multiple-way data augmentation	Pre-trained models (PTMs) to learn features, and a novel (L, 2) transfer feature learning algorithm was proposed to extract features	Private dataset from local hospitals		284 COVID-19 images, 281 community-acquired pneumonia images, 293 secondary pulmonary tuberculosis images, and 306 healthy control images.	On the test set, CCSHNet achieved sensitivities of four classes of 95.61%, 96.25%, 98.30%, and 97.86%. The precision values of four classes were 97.32%, 96.42%, 96.99%, and 97.38%. TheF1 scores of four classes were 96.46%, 96.33%, 97.64%, and 97.62%. The MA F1 score was 97.04%
(Demir et al., 2021) [20]	Automatic detection of COVID-19 from X-ray images	Deep LTSM	Marker-controlled watershed segmentation (MCWS)	20 convolutional layers	Public	Three classes: pneumonia, normal, and COVID-19. MCWS images were sized to 100 × 100 for the input layer	1061 CX images (361 COVID-19, 200 normal, and 500 pneumonia)	In 80% of tests, 100% performance was achieved for all aspects (accuracy, sensitivity, precision, and F-score)
(Sheykhivand et al., 2021) [21]	Automatic detection of COVID-19 from Chest images	Deep neural network	Generative Adversarial Networks (GANs) were used together with a fusion of the deep transfer learning and Long Short-Term Memory (LSTM) networks, without involving feature extraction/selection for classification of pneumonia.		Public four classes (healthy, COVID-19, bacterial, and viral)		Healthy: 2923 images; COVID-19:371; bacterial = 2778; viral = 2840	90% accuracy for all scenarios except one

**Table 2 diagnostics-12-01880-t002:** Different COVID-19 CXR datasets used in the investigated works.

N^o^	Ref.	Dataset Name	Nbr of Images	Resolution of Images	Include CT Images?	Observations
1.	[62]	Cohen Image Collection	315	4248 × 3480	No	Proposed image data are linked with clinical attributes
2.	[64]	COVIDx (COVID-19 CXR Dataset Initiative)	48	Various	-	-
3.	[65]	ActualMed COVID-19 CXR	13,975 CXR images	Various	No	Proposed COVID-Net improves the decision making of clinicians
4.	[66]	COVID-19 Radiography Database	2905	-	No	Specificity: 98.8%; sensitivity: 97.9%; precision: 97.95%; accuracy: 97.9%
5.	[67]	Japanese Society of Radiological Technology	105	4020 × 4892	No	ROC analysis gives Az values between 0.57 and 0.99
6.	[68]	CXR-8	112,120	1024 × 1024	No	High number of images gives better reults in ML algorithms
7.	[69]	SIRM COVID-19 Database	68	Various	Yes	-
8.	[70]	Radiopaedia.org	-	Various	Yes	Open dataset with increasing number of images
9.	[71]	ChexPert Dataset	224,316	320 × 320	No	Expert comparisons and uncertainty labels are considered
10.	[72]	Twitter COVID-19 CXR Dataset	135	2012 × 2012	No	-
11.	[73]	Pediatric Pneumonia CXR	5856	Various	No	Eight codes were proposed to evaluate this dataset
12.	[74,75]	OCT and Kaggle CXR images	5863	Various	Yes	OCT data are divided into training and testing sets with different patients
13.	[76]	Open-I Repository	-	Various	Yes	Statistical analysis of data is given

**Table 3 diagnostics-12-01880-t003:** Performance of reviewed detection models.

S.No.	Research	Accuracy (%)	F1 Score	AUC
1.	Covid-AID [77]	90.5	0.9230	0.99
2.	Deep Covid [78]	83	0.83	0.90
3.	Covid Caps [79]	95.7	-	0.97
4.	COVID Net [65]	93.3	-	-
5.	DeTrac [35]	95.12	-	-
6.	CheXNet [80]	97.8	97.8	-
7.	COVID-DA [81]	-	0.9298	0.985
8.	CoroNet [82]	93.5	93.51	-
9.	DenseNet-121 [54]	96.3	0.96	0.88
10.	ResNet-50 [83]	97.4	0.96	0.86
11.	Inception-V4 [45]	91.68	0.76	0.87
12.	Inception-ResNet-V2	89.45	0.84	0.86
13.	Xception [53]	81	0.80	0.88
14.	EfficientNet-B2	79	0.80	0.87
15.	ResNet-50 [8]	99.5	-	-
16.	ResNet50 and VGG16 [9]	99.87	-	0.83
17.	Transfer learning InceptionV3 [17]	99.49	0.85	-

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
