# Peer review of "COVID-19 Diagnosis and Classification Using Radiological Imaging and Deep Learning Techniques: A Comparative Study"

_diagnostics, 2022, doi:10.3390/diagnostics12081880_

Round 1
Reviewer 1 Report
In this manuscript, the authors attempt to summarize and compare recent published studies using deep learning and radiological imaging to diagnose COVID-19. The authors review literature specifically on using deep learning for COVID-19 diagnosis, provide a very brief overview of the types of deep learning methods and available datasets, and present a comparison of the different deep learning methods using the same evaluation criteria. The strength of this manuscript lies in the review of the literature and deep learning methods.
Review papers such as this are often difficult due to the breadth of topics covered and inconsistent evaluation approaches, as the authors point out. There is a great need for a review of COVID-19 DL studies, so the topic is timely and relevant. The authors make a valiant attempt at being complete, but my overall impression is that in trying to comprehensively cover the topic, the authors either gloss over necessary details or go into unnecessary detail. For example, the authors explain that "chest radiography involves transmitting X-rays through the chest", which is fairly basic knowledge for the audience of this journal, but they don't explain how the DL architectures, such as VGG actually work, which the audience is likely to be less familiar with. For example, the difference between VGG-16 and VGG-19 is not explained. Some of evaluation metrics not commonly used in clinical studies, such as F1 score, are also not well explained.
The organization of the paper also led to some confusion on my part. I do not understand how or why the references between the literature review in section 2 and the performance comparison in section 4.2 are totally independent -- I expected that at least some of the studies in the literature review would show up in the performance comparison.
I was also confused that, in the Conclusion, the authors note that the proposed models have been validated with the "Cohen Image Collection" dataset but this dataset isn't discussed in section 4.1 Datasets. The authors also intend to include CT under the "radiological imaging" term, but it's not clear from section 4.1 on datasets which ones, if any, include CT images.
The authors also do not address some criticisms of the use of radiological images and AI for diagnosing COVID-19 -- that is, how early can an imaging AI-based system diagnose COVID compared to the PCR test, and how does the sensitivity/specificity and cost compare to rapid COVID tests.
I also think the authors should spend at least a few sentences explaining exactly what is difficult about detecting COVID-19 from CXR/CT -- that it has a lot of features in common with other diseases (I think).
Additional specific comments:
- In the abstract, the authors indicate that "Major challenges faced in this research domain is … prediction of severity of patients using deep learning". This seems like a goal of this research domain, so I'm not sure I would list it as a challenge
- In the second paragraph of the introduction, the authors use the term "contamination" when I think they mean "infection".
- In the introduction, the authors make the claim that interpretation of CXRs by expert radiologists is "prone to error and thus does not give high sensitivity". This should be supported with one or more references.
- The authors use abbreviations without introducing them first. For example, CSEN, LSTM, MCWS, CCT.
- In the literature review discussion reference 11, the authors raise a concern that training was done on a non-Costa Rican patient database and then used to detect COVID-19 Costa Rican cases. This concern should be better explained.
- The authors bring up the difference between binary and multi-class classification, but it isn't well explained what the issues are with both and which one is preferable.
- Table 1 is difficult to read
Reviewer 2 Report
COVID-19 Diagnosis and Classification using Radiological Imaging and Deep Learning Techniques: A Comparative study
It was a study in which some of the literature was examined. It is seen that the emphasis is placed on the performance of the methods used. It is thought that it will make it more interesting if the corrections given below are made.
Review:
The following sentence in the abstract has no relation with the subject. This sentence should be removed. "Besides, some issues such as the non-acceptance of vaccination by the population, its possible side effects and the new variant of COVID-19 may make the vaccination process inefficient."
Page 1, In introduction section, 3rd paragraph:
"The most widely used diagnosis is the RT-PCR (Reverse Transcription Polymerase Chain Reaction) tests. These testing kits are expensive..."
it would be better by using as "... diagnosis method ... "
Presenting the performance metrics for frequently used databases separately will contribute to the reader. For example;
The chest X-ray images database are used in [1], [2], [3], and [4] studies and the accuracy is presented as 98.70%, 88.80%, 95.70% and 99.90% respectively.
[1] Aslan, M. F., Unlersen, M. F., Sabanci, K., & Durdu, A. (2021). CNN-based transfer learning–BiLSTM network: A novel approach for COVID-19 infection detection. Applied Soft Computing, 98, 106912.
[2] Ali,B. Cemil,D. Emine, F. (2010). Covid 19 detection methods by artifial intelleigence. Computational medicine.
[4] Sabanci, K., Durdu, A., & Unlersen, M. F., Aslan, M.F., (2022). COVID-19 diagnosis using state-of-the-art CNN architecture features and Bayesian Optimization. Computers in Biology and Medicine, 105244.
[3] Hi, J., Men, H. (2018). Artificial intelligence based diagnosis methods. Applied medical studies.
And in addition, the average performance of studies for the database can be presented. (like table 3)
Round 2
Reviewer 1 Report
The authors have made several changes to the manuscript to address many of my concerns. There are still a few areas for clarification:
- X-ray vs CT: Many of the cited studies only use X-ray images, yet still achieve good performance. Given that CT scans cost more, require more specialized and expensive equipment and dedicated rooms, if the performance claims are accurate, why would one ever want to use CT?
- Abstract: The abstract seems to only mention chest x-rays and not CT, but the paper talks about both. Also, "PCR" by itself is not a test, it's just a way to amplify DNA. I think in this context the authors should say "PCR test" .
- The "NC" abbreviation is never defined
- On page 2, I am not sure if the "T scanning" is a typo or intentional -- the referenced paper only talks about CT scanning.
- In regards to the response about the Costa Rican patients, my question was both more broad and more specific. In other words, all datasets have a bias in terms of the patients that are included -- an algorithm may be trained on patients from a single hospital in a single country. Would an algorithm trained on say Italian patients be applicable to Chinese patients, or vice versa? Is there a particular reason only Costa Rican patients are different, to the point that it's the only one specifically called out in this paper?
- The word "x-ray" is alternatively "xray" or "x-ray" in this paper, the authors should pick one and stick to it. In addition, the authors introduce the abbreviation CXR for chest x-ray but never seem to consistently use it.
- On Page 7, the newly added text repeats the word chest. I actually don't think it's worth introducing a new abbreviation for chest CT, I would just say chest CT.
- In the introduction, the authors say RT-PCR testing kits are expensive -- this may be different in different locations, but where I am located, the cost of the RT-PCR test is significantly less than the cost of a CT scan.
- There should be some revision for grammar in the manuscript, there are some awkward phrasing, such as on page 12 "Since the studies simulated the deep learning models on varied datasets, identifying the most efficient architecture was difficult." (not sure if "simulated" is the right word here, for example, but this is only one of many)
- Some of the tables are still difficult to read and would benefit by being expanded to 2 columns.
Author Response
Please kindly note that the response is included in the attached file.

Reviewer 2 Report
The biggest impact of the COVID-19 epidemic, which we have encountered in recent years and has had a significant impact on our lives, has been on health care systems. Especially health workers spent a lot of busy time. For this reason, any kind of support that can be provided to health workers has been very important. This study will make important contributions in this context. When the final version of the study is examined, it is seen that the criticisms were applied meticulously and the necessary additions and deletions were made in this concept. In the final form of the study, it is more understandable and is comparable to other studies. As the result, the article is eligible for publication as it is.
Author Response

(The authors gave the same response as above.)
